# Global, regional and national burden of injuries caused by fire, heat, and hot substances from 1990 to 2021

**Shi Huang**[1]*, **Hui-Zhen Lin**[2], **Xin Wei**[1]

**1** Department of Burns and Plastic Surgery, Affiliated Wuming Hospital of Guangxi Medical University, Nanning, China, **2** Clinical Laboratory Department, Affiliated Wuming Hospital of Guangxi Medical University, Nanning, China

☯ These authors contributed equally to this work: Shi Huang, Hui-Zhen Lin and Xin Wei contributed equally to this article.
\* huangshiwuming@163.com (SH)

## Abstract

### Background

Burn injuries, which are caused by fire, heat, and hot substances, are considered a chronic condition due to their long-term effects on the health of affected individuals. Moreover, burn injuries constitute a significant public health issue that cannot be overlooked within the global healthcare system.

### Materials and methods

This study comprehensively analyzed the burden of burn injuries, focusing on variations by Socio-Demographic Index (SDI) levels and Global Burden of Disease (GBD) regions, sex-based disparities, Frontier analysis, and future trend forecasting using the Bayesian Age-Period-Cohort (BAPC) model.

### Results

Compared to 1990, the global burden of burn injuries decreased in most regions worldwide in 2021. The majority of regions showed a negative Estimated Annual Percentage Change (EAPC), indicating a continuous decline in Age-Standardized Rate (ASR) annually. And the ASR decreased with the increasing SDI. Moreover, the forecasted trend from 2021 to 2030, indicates a continued decline with a well-fitted projection model.

### Conclusions

Despite the observed and projected decline in global ASR of burn, the burden of burn remains a significant concern that should not be underestimated.

**Data availability statement:** This study utilized data resources from the Global Burden of Disease database(https://ghdx.healthdata.org/gbd-2021), provided by the Institute for Health Metrics and Evaluation (IHME) at the University of Washington.

**Funding:** Self-funded research project of Health and Family Planning Commission of Guangxi Zhuang Autonomous Region(Z20180240). And the funders had no role in study design, data collection and analysis, decision to publish, or preparation of the manuscript.

**Competing interests:** The authors have declared that no competing interests exist.

# Background

In the realm of global public health, burn injuries constitute a significant and distinct segment [1]. Burn injuries are inherently acute conditions; however, in certain circumstances, the long-term sequelae of burn injuries may exhibit characteristics akin to those of chronic diseases [2]. Burn injuries not only damage skin tissues but also can lead to a range of complications [3]. At the same time, severe burn injuries have long-lasting effects on both the physical and psychological health of patients, thereby imposing a substantial disease burden [4]. Therefore, prompt and comprehensive management is essential for burn patients to mitigate the disease burden and improve outcomes [5]. However, owing to disparities in medical standards worldwide, not all burn patients have access to appropriate medical services. Consequently, epidemiological studies on burn injuries worldwide are of particular importance.

A recent epidemiological study indicate that in 2019, the global economic burden attributable to burn amounted to USD 11.2 billion, accounting for 0.09% of the global Gross Domestic Product (GDP) [6]. Meanwhile, a single-center epidemiological study in Southwest China reported a median medical costs of USD 29,152.54 for burn injuries patients in the region; another study conducted in the Netherlands reported an average total costs of EUR 26,540 per patient for burn injuries in that area [7,8]. Therefore, the economic burden imposed by burn injuries is substantial. Additionally, scholars utilized the Electronic Death Registry System (EDRS) to analyze the disease burden caused by burn injuries in southern Iran. The study results indicated a declining trend in crude and standardized mortality rates among burn injury patients in the region during the study period, with higher mortality rates observed in the 15–29 age group [9]. Overall, current epidemiological studies of burn injuries predominantly focus on national, provincial, multi-center, or single-center levels, with a notable deficiency in research on global burden of burn injuries.

Therefore, this study estimates the Global Burden of Disease (GBD) of burn using the GBD 2021 (https://ghdx.healthdata.org/gbd-2021). Despite previous reports on the GBD of burn injuries from 1990 to 2017, there remains a lack of comprehensive analysis based on the Socio-Demographic Index (SDI), sexes, and age groups for the current GBD of burn injuries. Additionally, future trends in GBD of burn injuries remain to be predicted [1]. Therefore, this study aims to examine the global burden of burn injuries caused by fire, heat, and hot substances, utilizing data from the GBD 2021 database. Specifically, the analysis focuses on the period from 1990 to 2021, while forecasting the burden for the period from 2021 to 2030. By providing a comprehensive epidemiological profile of burn injuries, this research seeks to offer health policymakers worldwide a robust evidence base to inform strategies aimed at reducing and preventing the burden of burn injuries.

# Materials and methods

## Data source

In the GBD 2021 study, the data from 56,640 data sources demonstrate extensiveness and inclusivity, achieved through the inclusion of 204 countries and territories,

as well as 811 subnational locations. The data sources encompass vital registration, verbal autopsies, censuses, disease surveillance systems, and cancer registries, and other relevant sources. The GBD 2021 study provides annual estimates of Incidence, Prevalence, Death, Disability-Adjusted Life Years (DALYs), Years of Life Lost (YLLs), and Years Lived with Disability (YLDs) from 1990 to 2021, stratified by age, sex, and region. This stratification ensures that that the results comprehensively reflect the health status of different regions and populations worldwide. Furthermore, through the analysis of the Socio-Demographic Index (SDI) and demographic data, the GBD 2021 has further enhanced the comprehensiveness and inclusivity of the data. This approach enables a deeper understanding of the relationship between development levels in different regions and health outcomes, thereby offering a comprehensive and in-depth perspective for global health research [10].

In this study, the data on burn injuries were sourced from the Global Health Data Exchange (GHDx) as part of the GBD 2021. The use of GBD 2021 data was approved by the Institutional Review Committee at the University of Washington and the Institute for Health Metrics and Evaluation (IHME). As the data are publicly accessible, there are no ethical concerns associated with this research [11].

### GBD injury classification

The GBD classification system categorizes the causes of injuries and the nature of those injuries. The external cause categories, or "codes", are used for events such as falls and road traffic accidents, as well as fire, heat, and hot substances. The injury nature codes classify injuries into 47 categories, quantifying various disabling outcomes for each injury cause according to the International Classification of Diseases (ICD), particularly the T20-T32 chapters of ICD-10 and the 900–999 codes of ICD-9. The latest GBD literature provides more information on the nature of injury groups. Within the GBD 2021 dataset, burn injuries are defined as a group of diseases characterized by injuries to the skin or other tissues caused by flames, hot liquids, hot gases, chemicals, or other sources of heat [1,12].

### Data collection and processing

We describe the GBD of injuries caused by fire, heat, and hot substances between 1990 and 2021 by considering various metrics, including Number of Incidence, Prevalence, Death; DALYs, YLLs, and YLDs, as well as Age-Standardized Rate (ASR) and Estimated Annual Percentage Changes (EAPC) [13]. Incidence is defined as the number of new cases in a population over a specific time frame and reflects how often the disease occurs in the population. Prevalence refers to the number of existing cases of a specific disease in a specific population at a specific point in time or over a period of time and includes two forms: point prevalence and period prevalence. Death rate is defined as the number of deaths due to a specific cause within a certain period of time. DALYs represent the total healthy life years lost from the onset of illness to death, including both YLLs and YLDs.

ASR stands for the age-standardized rate, which makes the burden of burn injuries more comparable across different regions, sexes, and age groups by accounting for age distribution. EAPC can be used to assess the average annual growth rate of a certain indicator over a specific time period. In the calculation formula for EAPC mentioned in S1 File, if $\beta$ is less than 0, then $\exp(\beta)$ is less than 1, indicating that the ASR shows an overall decreasing trend during 1990–2021. Conversely, if $\beta$ is greater than 0, then $\exp(\beta)$ is greater than 1, indicating that the ASR shows an overall increasing trend during 1990–2021 (S1 File, Chapter 1).

In this study, we employed uncertainty interval (UI) and confidence interval (CI) to characterize the margins of error for Number, ASR, and EAPC. The UI for Number and ASR provides a quantification of the uncertainty surrounding these estimates. The calculation of UI is based on the variability of the statistical models and the inherent uncertainty of the data, reflecting the credibility and precision of the estimates, which are typically provided by the GBD database. Conversely, the CI for EAPC offers a quantification of the uncertainty associated with the EAPC estimates. The computation of CI is based on the standard error of the regression analysis and the t-distribution (or other distributions, contingent upon assumptions

regarding sample size and distribution) to determine the confidence interval for the regression coefficient β, thereby facilitating the calculation of the EAPC confidence interval [14,15].

To obtain and analyze the Number, ASR and EAPC of Incidence, Death, Prevalence, DALYs, YLLs, YLDs for burn injuries, we utilized R programming packages, including "dplyr" and "ggplot2" for the analysis. Additionally, based on 22 GBD regions and 204 countries and territories, this study conducted a detailed comparative analysis of the burden caused by burn injuries across different SDI groups, countries and sexes.

Subsequently, Frontier analysis was conducted based on the burden of burn injuries from 1990 to 2021 across various countries. In this context, Frontier analysis is a methodology employed to assess the disparities between different countries or regions and the global optimum in health performance, which is denoted as the "Frontier". The "Frontier" refers to the countries or regions that have achieved the lowest burden of burn injuries at a given SDI level, thereby exemplifying best practices in health performance. The objective of Frontier analysis is to identify and quantify the gaps between other countries or regions and these best-practice countries, thereby uncovering potential opportunities for health improvement. Within Frontier analysis, the "Effective Difference" (EF) is commonly calculated. It denotes the distance between a country or region and the Frontier, reflecting the discrepancy between the actual burn burden of the country or region and the lowest attainable burden. A larger EF indicates a greater untapped potential for health improvement within the country or region. [15–19] (S1 File, Chapter 2).

Finally, the study utilized a Bayesian Age-Period-Cohort (BAPC) model to forecast the burden of burn injuries from 2021 to 2030 [20]. In the BAPC model, unknown parameters are treated as random variables with prior distributions, and observational data from samples are combined with prior knowledge to estimate the posterior distribution. Assuming that the effects between adjacent time periods are similar, we employed a second-order random walk prior based on the linear time trend assumption for period effects to adjust for over-dispersion [21,22]. We organized data on annual Prevalence rates, Death rates, and DALYs from 1990 to 2021 into 31 consecutive single-year periods and 17 consecutive age groups (0–9 years, 10–14 years, …, 80–84 years, >85 years). Subsequently, corresponding population data were organized into 41 consecutive single-year periods from 1990 to 2030 [23] (S1 File, Chapter 3). All analyses in this study were conducted using R (version 4.2.3).

## Results

### Global and regional burden of burn injuries from 1990 to 2021

**DALYs, YLLs, and YLDs from 1990 to 2021.** Compared to 1990, there was a decrease in number and ASR of DALYs, YLLs, and YLDs globally due to burn injuries in 2021. Globally, the EAPC for DALYs, YLLs, and YLDs were -2.26 ([95% CI] -2.36 to -2.15), -2.36 ([95% CI] -2.49 to -2.24), and -2.02 ([95% CI] -2.10 to -1.94), respectively. The most significant decline in ASR of DALYs was observed in East Asia with an EAPC of -4.65 ([95% CI] -4.78 to -4.52); for ASR of YLLs, Australasia showed the largest decline with an EAPC of -5.29 ([95% CI] -6.01 to -4.57); and for ASR of YLDs, Tropical Latin America exhibited the most notable decrease with an EAPC of -3.56 ([95% CI] -3.83 to -3.28). Notably, despite an upward trend in number of DALYs in regions like Australasia and Oceania, their negative EAPC values indicate a decreasing ASR of DALYs in 2021 compared to 1990. In the broader trend of declining disease burden, YLDs in Oceania showed an increase in both number and ASR, with an EAPC of 0.11 ([95% CI] 0.01 to 0.21), suggesting a rise in disease burden in this region from 1990 to 2021 (Tables 1). Additionally, Figs 1 reveals that the global burden of burn, as reflected in DALYs, YLLs, and YLDs, has shown a continuous decline from 1990 to 2021. Furthermore, the number and ASR are higher for males compared to females (Figs 1).

### Incidence, prevalence and death from 1990 to 2021

Compared to 1990, there was a decrease in both number and Age-Standardized Incidence Rate (ASIR) and Age-Standardized Death Rate (ASDR) caused by burn injuries in 2021. The EAPC for Incidence and Death were -1.33 ([95% CI]

**Table 1. Burden of injuries caused by fire, heat, and substances in Disability-Adjusted Life Years, Years of Life Lost and Years Lived with Disability between 1990 and 2021.**

| | 1990 Number | 2021 Number | 1990 | 2021 | Estimated Annual Percent Change |
|---|---|---|---|---|---|
| | (95% UI) | (95% UI) | Age-Standardized Rate | Age-Standardized Rate | (95% CI) |
| | | | (95% UI) | (95% UI) | |
| **DALYs** | | | | | |
| Global | 11458000.8 | 8470611.5 | 211.5 | 108.6 | -2.26 |
| | (9652567.6-12962184.7) | (6900996.8-10126587.0) | (179.4-240.7) | (87.6-129.2) | (-2.36--2.15) |
| Andean Latin America | 117881.4 | 72353.7 | 304.7 | 109.8 | -3.52 |
| | (98944.7-136963.9) | (57030.3-92373.9) | (253.4-357.4) | (86.8-140.1) | (-3.66--3.39) |
| Australasia | 22832.5 | 23457.4 | 108.5 | 64.1 | -1.88 |
| | (15596.4-33358.4) | (13834.9-37290.8) | (75.3-156.5) | (38.5-100.1) | (-2.10--1.66) |
| Caribbean | 137122.3 | 106777.2 | 377.2 | 228.7 | -1.49 |
| | (109530.5-164727.4) | (83775.8-131981.7) | (301.6-450.4) | (178.1-282.1) | (-1.66--1.31) |
| Central Asia | 342418 | 170279.2 | 476.2 | 175.2 | -3.79 |
| | (310525.4-378650.1) | (143614.8-205370.5) | (427.9-533.1) | (147.6-211.4) | (-4.10--3.48) |
| Central Europe | 236967.1 | 125932.3 | 183.8 | 85.7 | -2.69 |
| | (192378.6-290290.8) | (95279.4-168748.9) | (150.1-223.5) | (65.3-113.9) | (-2.80--2.58) |
| Central Latin America | 571857 | 343758.5 | 349.8 | 133.1 | -2.64 |
| | (449339.2-707118.2) | (262601.3-446266.9) | (273.3-432.3) | (102.0-172.5) | (-2.86--2.42) |
| Central Sub-Saharan Africa | 290221.3 | 289505.2 | 444.9 | 245.2 | -1.85 |
| | (223428.5-358491.2) | (211035.3-547644.8) | (352.8-539.4) | (183.9-441.4) | (-1.99--1.72) |
| East Asia | 1539552.1 | 556614.9 | 134.7 | 33.8 | -4.65 |
| | (1258711.4-1800055.6) | (443425.2-701185.5) | (109.9-157.2) | (27.2-41.7) | (-4.78--4.52) |
| Eastern Europe | 692765.9 | 376935.6 | 299.4 | 147.2 | -3.25 |
| | (608035-801016.5) | (318820.5-457298) | (265.0-343.0) | (125.6-176.7) | (-4.13--2.35) |
| Eastern Sub-Saharan Africa | 1045740.7 | 965834.4 | 483.4 | 256.1 | -2.05 |
| | (831580.8-1236161.7) | (758274.4-1271632.1) | (407.8-575.2) | (207.1-329.2) | (-2.08--2.01) |
| High-income Asia Pacific | 234653.5 | 164526.1 | 129.1 | 62.4 | -2.56 |
| | (170732.2-326391.4) | (108297.8-249347.4) | (96.3-176.1) | (40.3-94.2) | (-2.73--2.40) |
| High-income North America | 431911.8 | 268143.6 | 152.6 | 60.9 | -2.88 |
| | (347913.7-558836.7) | (196122.1-377814.8) | (125.8-192.5) | (45.9-83.7) | (-3.11--2.65) |
| North Africa and Middle East | 1249670.8 | 662852.4 | 333.4 | 106.6 | -3.91 |
| | (762888.1-1511094.4) | (544968.7-797058.9) | (206.4-398.3) | (87.5-127.9) | (-4.04--3.77) |
| Oceania | 19782.6 | 37545 | 290.8 | 260.6 | -0.37 |
| | (13482.3-27162.5) | (25836.1-52946.9) | (212.9-377.2) | (190.3-354.2) | (-0.51--0.24) |
| South Asia | 2280325.2 | 2010713.1 | 201.2 | 108.1 | -2.07 |
| | (1849246.9-2605560.3) | (1595304.7-2287816.9) | (165.5-228.4) | (86.4-123.0) | (-2.19--1.95) |
| Southeast Asia | 557096.6 | 423811.5 | 118.7 | 60.4 | -2.11 |
| | (451290.6-639945.3) | (351689.6-510628.6) | (98.1-136.1) | (50.2-72.5) | (-2.16--2.06) |
| Southern Latin America | 131330.3 | 109724.8 | 269.9 | 149.5 | -1.73 |
| | (100615.3-172711.5) | (74274.3-160577) | (206.0-355.9) | (102.4-216.7) | (-1.8909--1.56) |
| Southern Sub-Saharan Africa | 228933.8 | 230538.2 | 443.3 | 290.4 | -1.17 |
| | (192082.6-263163.1) | (195875.8-268472.4) | (377.2-511.6) | (247.7-337.6) | (-1.40--0.95) |
| Tropical Latin America | 281448.6 | 168305.7 | 192.6 | 69.2 | -3.72 |
| | (236435-328344.1) | (130615.1-212814.1) | (160.9-226.8) | (54.0-87.1) | (-3.91--3.52) |

*(Continued)*

| | 1990 Number | 2021 Number | 1990 | 2021 | Estimated Annual Percent Change |
|---|---|---|---|---|---|
| | (95% UI) | (95% UI) | Age-Standardized Rate | Age-Standardized Rate | (95% CI) |
| | | | (95% UI) | (95% UI) | |
| Western Europe | 394466.8 | 272014.2 | 95.3 | 49.1 | -2.2 |
| | (286477.3-548874.4) | (175691.7-411877.6) | (71.0-129.9) | (32.1-73.8) | (-2.26--2.13) |
| Western Sub-Saharan Africa | 651022.7 | 1090988.3 | 275.3 | 213.3 | -0.81 |
| | (476530.7-787691.8) | (539397.7-1475115.4) | (210.0-327.3) | (115.2-284.8) | (-1.04--0.57) |
| **YLLs** | | | | | |
| Global | 8220450.8 | 5573490.4 | 146.8 | 73.4 | -2.36 |
| | (6696932.6-9152343.1) | (4168061.5-6706244.0) | (120.5-162.4) | (54.0-89.2) | (-2.49--2.24) |
| Andean Latin America | 57856.2 | 22589 | 131.5 | 35 | -4.69 |
| | (47267.2-65985.4) | (18197.5-28855.7) | (108.3-148.3) | (28.2-44.7) | (-5.03--4.36) |
| Australasia | 6964.4 | 2688.6 | 35.7 | 8.4 | -5.29 |
| | (6750.0-7207.7) | (2563.9-2816.2) | (34.5-37.0) | (7.9-8.8) | (-6.01--4.57) |
| Caribbean | 73104.5 | 39443.8 | 190.8 | 91 | -2.16 |
| | (53624.9-94278.3) | (28329.9-52618.7) | (142.9-243.9 | (63.6-123.0) | (-2.46--1.86) |
| Central Asia | 239581.1 | 89369.2 | 312.2 | 91.9 | -4.59 |
| | (223024.5-255329.7) | (79553.1-104245.0) | (292.7-330.2) | (81.9-106.9) | (-5.00--4.18) |
| Central Europe | 97125.7 | 50961.7 | 79.9 | 35 | -3.14 |
| | (94586.5-100389.8) | (47614.4-54324.4) | (77.6-82.7) | (32.5-37.5) | (-3.35--2.93) |
| Central Latin America | 137215 | 68055.9 | 76.8 | 27.2 | -3.4 |
| | (130965.5-145168.5) | (60680.1-77390.7) | (74.0-80.4) | (24.1-31.1) | (-3.75--3.05) |
| Central Sub-Saharan Africa | 257311.2 | 235984.3 | 373.4 | 198.4 | -1.96 |
| | (193361.3-320592.3) | (160862.9-501044.6) | (287.2-455.8) | (138.8-391.7) | (-2.11--1.82) |
| East Asia | 1166581.4 | 299191.6 | 103.7 | 19.7 | -5.57 |
| | (894716.8-1360618) | (229993.4-363949.8) | (80.3-120.8) | (15.6-23.7) | (-5.72--5.42) |
| Eastern Europe | 437703 | 231377.1 | 197.6 | 93 | -3.59 |
| | (428782.1-446284.3) | (213525.9-249871.7) | (193.4-202.2) | (86.3-100.2) | (-4.66--2.50) |
| Eastern Sub-Saharan Africa | 874456.3 | 705648.8 | 376.4 | 184.5 | -2.3 |
| | (651482.7-1046487.2) | (514918.9-991547.3) | (300.5-455.3) | (142.0-252.1) | (-2.35--2.25) |
| High-income Asia Pacific | 93755.8 | 41879.4 | 56.4 | 15.4 | -4.51 |
| | (85584.1-100822.9) | (38120.4-45216.0) | (51.4-61.3) | (14.5-17.0) | (-4.85--4.17) |
| High-income North America | 245430.1 | 108416.6 | 92.9 | 27.3 | -3.66 |
| | (242083.1-248807.1) | (104202.8-111365.9) | (91.8-94.3) | (26.3-28.2) | (-3.91--3.40) |
| North Africa and Middle East | 1146712.4 | 568637.6 | 296.4 | 91.3 | -4.04 |
| | (653758.9-1406058.2) | (439721.3-689963.1) | (168.3-358.6) | (70.5-110.2) | (-4.18--3.89) |
| Oceania | 15207.8 | 26831.4 | 210.7 | 177.3 | -0.58 |
| | (9038.1-21996.5) | (16370.5-41844.4) | (139.7-292.7) | (116.0-267.3) | (-0.74--0.41) |
| South Asia | 1962235 | 1659783.2 | 167.7 | 89.1 | -2.09 |
| | (1539353.6-2254002.2) | (1221749.2-1908202.8) | (132.2-190.9) | (65.8-102.6) | (-2.20--1.98) |
| Southeast Asia | 356202.3 | 206829.1 | 70.5 | 31.1 | -2.62 |
| | (268076.1-415241.7) | (163463.9-243703.5) | (54.7-80.7) | (24.2-36.8) | (-2.69--2.56) |
| Southern Latin America | 55615.3 | 28757.1 | 113.1 | 41.6 | -2.98 |
| | (54397.6-56788.3) | (27513.7-30151.7) | (110.7-115.4) | (39.5-43.9) | (-3.26--2.71) |

*(Continued)*

**Table 1.** (Continued)

| | 1990 Number | 2021 Number | 1990 | 2021 | Estimated Annual Percent Change |
|---|---|---|---|---|---|
| | (95% UI) | (95% UI) | Age-Standardized Rate | Age-Standardized Rate | (95% CI) |
| | | | (95% UI) | (95% UI) | |
| Southern Sub-Saharan Africa | 183518 | 184719.7 | 340.2 | 232.6 | -0.93 |
| | (146952.2-211916.0) | (151803.9-217519.3) | (274.7-394.3) | (193.3-272.9) | (-1.23--0.62) |
| Tropical Latin America | 119496.4 | 51880 | 78.3 | 22.2 | -4.01 |
| | (112039.5-126935.8) | (49385.4-54347.1) | (74.1-82.5) | (20.8458-23.3967) | (-4.29--3.72) |
| Western Europe | 157611.5 | 62825.2 | 41.8 | 11.7 | -4.12 |
| | (154750.7-159605.5) | (59367.2-65023.4) | (41.2-42.4) | (11.3-12.1) | (-4.26--3.98) |
| Western Sub-Saharan Africa | 536767.4 | 887621.3 | 204.4 | 163.2 | -0.72 |
| | (362055.8-674589.1) | (333296.1-1241130.5) | (138.8-252.1) | (65.2-228.5) | (-1.01--0.43) |
| **YLDs** | | | | | |
| Global | 3237550 | 2897121.1 | 64.7 | 35.1 | -2.02 |
| | (2241786.2-4411065.7) | (1941036.7-4106470.5) | (44.5-88.6) | (23.6-49.6) | (-2.10--1.94) |
| Andean Latin America | 60025.2 | 49764.7 | 173.2 | 74.8 | -2.79 |
| | (43167.2-77905.8) | (34199.5-68668.0) | (124.1-223.6) | (51.5-103.4) | (-2.84--2.74) |
| Australasia | 15868.1 | 20768.9 | 72.9 | 55.7 | -0.78 |
| | (8662.2-26339.9) | (11164.6-34611.5) | (39.7-120.6) | (29.9-91.7) | (-0.84--0.73) |
| Caribbean | 64017.8 | 67333.4 | 186.4 | 137.7 | -0.94 |
| | (46034.8-82458.4) | (48282.7-89496.2) | (134.2-239.2) | (99.6-182.0) | (-1.03--0.84) |
| Central Asia | 102836.9 | 80910 | 164 | 83.3 | -2.5 |
| | (74317.7-135436.3) | (55581.9-113003.4) | (117.5071-218.9) | (57.2-116.2) | (-2.69--2.30) |
| Central Europe | 139841.4 | 74970.6 | 103.9 | 50.7 | -2.32 |
| | (95933.2-194843.3) | (43857.6-116363.5) | (71.1-144.8) | (29.8-78.0) | (-2.41--2.24) |
| Central Latin America | 434641.9 | 275702.6 | 273 | 105.9 | -2.43 |
| | (308911.9-565901.7) | (194499.5-376507.1) | (196.5-355.1) | (74.9-144.7) | (-2.72--2.14) |
| Central Sub-Saharan Africa | 32910.1 | 53520.9 | 71.5 | 46.8 | -1.32 |
| | (23860.3-42822.3) | (38508.3-68810.6) | (52.2-91.1) | (33.7-60.7) | (-1.47--1.17) |
| East Asia | 372970.7 | 257423.3 | 31.1 | 14.1 | -2.62 |
| | (263363.6-493118.0) | (150734.8-396753.5) | (21.8-41.2) | (8.3-21.7) | (-2.72--2.51) |
| Eastern Europe | 255062.9 | 145558.6 | 101.8 | 54.2 | -2.4 |
| | (172242.0-364212.9) | (88320.1-222480.4) | (69.0-145.0) | (33.0-82.6) | (-2.82--1.99) |
| Eastern Sub-Saharan Africa | 171284.4 | 260185.6 | 107 | 71.5 | -1.25 |
| | (122697.3-222030.7) | (186869.4-336319.0) | (77.1-136.6) | (51.0-93.1) | (-1.32--1.18) |
| High-income Asia Pacific | 140897.7 | 122646.7 | 72.8 | 47 | -1.52 |
| | (78120.0-232579.8) | (66555.5-207622.3) | (40.4-119.5) | (25.3-78.6) | (-1.63--1.40) |
| High-income North America | 186481.7 | 159726.9 | 59.7 | 33.6 | -1.98 |
| | (102283.0-314999.6) | (87339.3-271133.4) | (32.7-100.3) | (18.3-57.1) | (-2.30--1.66) |
| North Africa and Middle East | 102958.4 | 94214.8 | 37 | 15.3 | -3 |
| | (72623.8-135034.8) | (62432.3-134891.0) | (25.9-49.2) | (10.0-22.0) | (-3.11--2.89) |
| Oceania | 4574.8 | 10713.6 | 80.1 | 83.3 | 0.11 |
| | (3315.4-5929.0) | (7799.5-13900.3) | (58.2-104.0) | (61.1-107.5) | (0.01-0.21) |

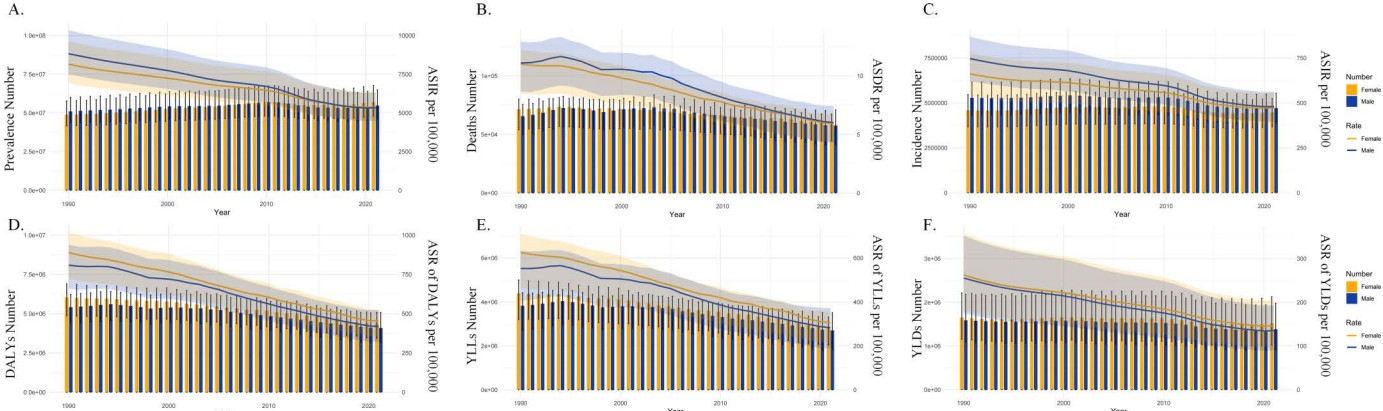

**Fig 1. Graph of successive trends in the burden of injury caused by fire, heat and hot substances between 1990 and 2021.** A-F represent the changes in the burden of Prevalence, Death, Incidence, Disability-Adjusted Life Years, Years of Life Lost and Years Lived with Disability of burn, respectively.

-1.42 to -1.23) and -2.18 ([95% CI] -2.33 to -2.03), respectively. However, in Oceania, both the number and ASIR increased, with an EAPC of 0.18 ([95% CI] 0.10 to 0.26). In contrast, Prevalence exhibited a slightly different trend compared to Incidence and Death. Globally, Age-Standardized Prevalence Rate (ASPR) decreased from 2117.3 cases per 100,000 population in 1990 to 1334.9 cases per 100,000 population in 2021, with an EAPC of -1.54 ([95% CI] -1.62 to -1.47), while number showed an increasing trend. However, in Oceania, both the number and ASIR and ASPR showed an upward trend, with EAPCs of 0.18 ([95% CI] 0.10 to 0.26) and 0.084 ([95% CI] -0.003 to 0.170), respectively (Tables 2). Additionally, Figs 1 reveals that the global burden of the disease, as reflected in Prevalence, Incidence, and Death, has shown a continuous decline from 1990 to 2021. Furthermore, the number and ASR are higher for males compared to females

## Comparison of burn injuries by sexes from 1990 to 2021

In Figs 1, the global trend of the disease burden over time, showing a continuous decline in these six indicators from 1990 to 2021. Moreover, the Number and ASR of DALYs YLLs and YLDs for females are higher than those for males. The differences between sexes in terms of Prevalence, Incidence, and Death are inversely related.

Specifically, comparison of GBD due to burn injuries by sexes globally from 1990 to 2021 revealed that on a global level, ASR of DALYs for females exceeded that for males. Additionally, ASR of YLDs, YLLs, and Prevalence were higher in females compared to males. Conversely, ASIR and ASDR were higher in males. (S2 File)

Furthermore, for 2021, this study illustrated sexes disparities in burn injuries. In high and middle SDI regions such as Central Asia, Oceania, and Australasia, males exhibited higher burn injuries burden compared to females across six metrics. Conversely, in South Asia, females consistently showed higher burn injuries burden than males (S3 File). In general, the global burden of burn injuries are more severe for males.

## Global, regional and national burn injuries burden based on SDI

High ASRs for Incidence (Figs 2A), Prevalence (Figs 2B) and YLDs (Figs 2F) are more prevalent in middle to high SDI regions such as the Americas, Europe, and Australia. Conversely, high ASRs for Death (Figs 2C), DALYs (Figs 2D), and YLLs (Figs 2E) are predominantly concentrated in lower SDI regions such as Africa, Oceania, and Eastern Europe. Furthermore, from 1990 to 2021, overall ASRs for Incidence, Prevalence, Death, DALYs, YLLs and YLDs generally decrease with increasing SDI (Figs 2A-F). Additionally, regions with middle to high SDI levels exhibit a more pronounced disease burden from burn injuries. Furthermore, the burden of burn injuries in 204 countries decreases with the increasing SDI (S1 Fig).

                                                                                 

**Table 2. Burden of injuries caused by fire, heat, and substances in Incidence, Prevalence and Death between 1990 and 2021.**

| | 1990 Number | 2021 Number | 1990 | 2021 | Estimated Annual Percent Change |
|---|---|---|---|---|---|
| | (95% UI) | (95% UI) | Age-Standardized Rate | Age-Standardized Rate | (95% CI) |
| | | | (95% UI) | (95% UI) | |
| **Incidence** | | | | | |
| Global | 9862413.1 | 9198163.9 | 176.1 | 118.8 | -1.33 |
| | (8005030.5-11572267.1) | (7592028.2-10784349.7) | (145.0-205.3) | (97.5-139.4) | (-1.42--1.23) |
| Andean Latin America | 150399.8 | 170721.4 | 343.1 | 255.2 | -1.03 |
| | (120257.8-180234.5) | (135990.8-205389.3) | (282.9-403.7) | (202.7-307.7) | (-1.06--1.00) |
| Australasia | 66853.7 | 68420 | 344.8 | 258.1 | -0.85 |
| | (53651.2-78523.4) | (55328.9-79773.9) | (275.8-406.6) | (204.2-307.5) | (-0.90--0.80) |
| Caribbean | 167183.7 | 166589.1 | 437.9 | 373.8 | -0.42 |
| | (134531.6-199733.7) | (135600.3-197178.1) | (356.5-521.6) | (301.0-446.4) | (-0.56--0.28) |
| Central Asia | 339575.1 | 279818.9 | 453.4 | 289.5 | -1.71 |
| | (286508.9-386986.1) | (231899.3-323856) | (384.6-512.0) | (241.5-333.5) | (-1.83--1.60) |
| Central Europe | 439988.8 | 265395.9 | 365.1 | 269.4 | -0.99 |
| | (363819.9-518194.7) | (220254.8-307250.4) | (300.2-432.4) | (215.0-320.8) | (-1.05--0.93) |
| Central Latin America | 1079802 | 780688.4 | 533.1 | 329.7 | -1.03 |
| | (845728.8-1354423.8) | (605576.9-978321.8) | (423.7-659.1) | (256.9-417.8) | (-1.31--0.75) |
| Central Sub-Saharan Africa | 84386. 3 | 162879.2 | 138.2 | 107.6 | -0.84 |
| | (64817.2-103454.6) | (126942.2-200895.3) | (110.8-163.5) | (87.1-128.7) | (-0.92--0.75) |
| East Asia | 1015437.2 | 890129.9 | 77.5 | 68.4 | -0.2 |
| | (771715.8-1247848.9) | (700328.5-1067688.3) | (59.9-94.5) | (51.7-84.1) | (-0.34--0.05) |
| Eastern Europe | 914823.7 | 563533.1 | 411.6 | 284.9 | -1.49 |
| | (771107.8-1057198.1) | (479300.6-645859.0) | (344.8-474.8) | (239.0-328.1) | (-1.87--1.10) |
| Eastern Sub-Saharan Africa | 468720.5 | 808080.3 | 213.4 | 164.8 | -0.95 |
| | (358005.9-583688.5) | (625516.7-990740.9) | (169.3-255.5) | (131.9-196.3) | (-1.02--0.88) |
| High-income Asia Pacific | 545881.5 | 316483.8 | 328.5 | 212 | -1.56 |
| | (452659.8-634318.9) | (260308.1-366897.5) | (270.6-381.1) | (167.9-253.3) | (-1.68--1.44) |
| High-income North America | 770785.3 | 548639.2 | 275.8 | 155.1 | -2 |
| | (636852.8-900869.8) | (451482.5-639920.1) | (227.1-322.0) | (126.4-183.6) | (-2.3--1.69) |
| North Africa and Middle East | 347901.7 | 379764.5 | 92.1 | 58 | -1.62 |
| | (276129.5-418676.3) | (309522.2-449067.3) | (73.5-109.2) | (47.4-68.7) | (-1.72--1.51) |
| Oceania | 13269.2 | 29708.1 | 180.5 | 196.2 | 0.18 |
| | (10723.1-15842.1) | (24306.8-34719.7) | (147.3-213.9) | (162.0-228.5) | (0.10-0.26) |
| South Asia | 825516.7 | 1056121.8 | 69.5 | 53.2 | -0.97 |
| | (641508.3-1000587) | (846499.8-1264030.2) | (54.6-83.7) | (42.8-63.1) | (-1.1--0.83) |
| Southeast Asia | 564535.4 | 640766.6 | 110.2 | 90.9 | -0.51 |
| | (435382.7-687655.9) | (505978.6-763168.7) | (87.1-131.5) | (71.4-108.6) | (-0.58--0.43) |
| Southern Latin America | 318432. 9 | 302325.7 | 630.9 | 484.3 | -0.7 |
| | (262522.9-370933.1) | (253496.5-347483) | (522.3-732.7) | (401.4-560.6) | (-0.85--0.55) |
| Southern Sub-Saharan Africa | 151293.1 | 158732 | 263.6 | 187.4 | -1.27 |
| | (122558.6-177655.2) | (132410.5-182981.6) | (216.5-308.6) | (157.3-215.7) | (-1.40--1.13) |
| Tropical Latin America | 462476.3 | 351867.2 | 277.2 | 161.3 | -2.51 |
| | (367612.1-561542) | (291760.9-410080.9) | (225.8-329.4) | (132.5-189.6) | (-2.81--2.21) |

*(Continued)*

| | 1990 Number | 2021 Number | 1990 | 2021 | Estimated Annual Percent Change |
| --- | --- | --- | --- | --- | --- |
| | (95% UI) | (95% UI) | Age-Standardized Rate | Age-Standardized Rate | (95% CI) |
| | | | (95% UI) | (95% UI) | |
| Western Europe | 827400.6 | 597171.2 | 244.9 | 172.7 | -1.2 |
| | (688169.4-977977.4) | (493834.0-694148.5) | (199.4-289.4) | (137.5-206.7) | (-1.23--1.17) |
| Western Sub-Saharan Africa | 307749.9 | 660328.3 | 141.4 | 119.5 | -0.56 |
| | (234817.5-385415.8) | (516317.0-806461.0) | (111.8-170.4) | (97.2-142.4) | (-0.65--0.48) |
| **Deaths** | | | | | |
| Global | 137006.9 | 117406 | 2.7 | 1.5 | -2.18 |
| | (113892.7-150141.8) | (92583.4-135183.8) | (2.3-3.0) | (1.2-1.7) | (-2.33--2.03) |
| Andean Latin America | 860.7 | 503.2 | 2.4 | 0.8 | -3.98 |
| | (714-966.7) | (413.7-621.6) | (2.0-2.7) | (0.7-1.0) | (-4.33--3.64) |
| Australasia | 156.9 | 82.7 | 0.74 | 0.19 | -4.98 |
| | (151.3-162.5) | (76-87.5) | (0.72-0.77) | (0.18-0.2) | (-5.66--4.29) |
| Caribbean | 1088.4 | 724.3 | 3.1 | 1.5 | -2.15 |
| | (856.2-1345.7) | (567.8-909.6) | (2.6-3.8) | (1.2-2.0) | (-2.43--1.87) |
| Central Asia | 3788.5 | 1745.3 | 5.6 | 1.9 | -4.114 |
| | (3563.8-3988.9) | (1562.1-1982) | (5.3-5.9) | (1.7-2.1) | (-4.52--3.71) |
| Central Europe | 2375.7 | 1735.6 | 1.8 | 0.9 | -2.53 |
| | (2316.9-2444.0) | (1616.7-1844.1) | (1.7 -1.9) | (0.8-1.0) | (-2.74--2.32) |
| Central Latin America | 2190.4 | 1619.7 | 1.57 | 0.6 | -2.96 |
| | (2115.7-2282.2) | (1462.7-1806.5) | (1.52-1.61) | (0.5-0.7) | (-3.26--2.66) |
| Central Sub-Saharan Africa | 3759.2 | 4390.6 | 8.5 | 5.8 | -1.24 |
| | (2864.0-4640.9) | (3065.7-8818.5) | (6.7-10.7) | (4.1-9.9) | (-1.34--1.14) |
| East Asia | 20038.2 | 10862.5 | 2.2 | 0.6 | -4.09 |
| | (15413.6-23117.5) | (8218.4-13220.4) | (1.7-2.5) | (0.5-0.8) | (-4.24--3.94) |
| Eastern Europe | 9682.7 | 6834.6 | 4.06 | 2.4 | -2.99 |
| | (9473.1-9841.3) | (6297.8-7374.2) | (3.97-4.13) | (2.2-2.5) | (-4.09--1.89) |
| Eastern Sub-Saharan Africa | 13047.7 | 12907 | 8.8 | 5.3 | -1.74 |
| | (10330.9-15595.8) | (9919.3-17650.5) | (7.3-11.1) | (4.2-6.8) | (-1.78--1.70) |
| High-income Asia Pacific | 2367.4 | 1992.9 | 1.3 | 0.48 | -3.645 |
| | (2209.9-2563.1) | (1712.9-2163.4) | (1.2-1.4) | (0.43-0.51) | (-3.93--3.35) |
| High-income North America | 5185.7 | 3343.4 | 1.76 | 0.65 | -2.89 |
| | (5033.8-5282.4) | (3132.3-3459.2) | (1.72-1.79) | (0.62-0.67) | (-3.11--2.66) |
| North Africa and Middle East | 17176.7 | 10837.7 | 5.4 | 1.9 | -3.62 |
| | (9739.5-20777.3) | (8509.5-12895.7) | (3.1-6.5) | (1.5-2.3) | (-3.78--3.47) |
| Oceania | 235.3 | 426.3 | 4 | 3.3 | -0.66 |
| | (155.7-326.6) | (289-636.5) | (3.0-5.4) | (2.4-4.8) | (-0.81--0.51) |
| South Asia | 31427.1 | 32843 | 3.2 | 1.9 | -1.72 |
| | (24839.6-35707.2) | (24688.1-37704.1) | (2.6-3.7) | (1.5-2.2) | (-1.79--1.64) |
| Southeast Asia | 5402.5 | 4355.8 | 1.3 | 0.7 | -2.07 |
| | (4316.4-6173.2) | (3523.8-5060.8) | (1.0-1.4) | (0.5-0.8) | (-2.14--1.99) |
| Southern Latin America | 1216.1 | 824.2 | 2.6 | 1 | -2.67 |
| | (1189.3-1244.4) | (777.6-865.1) | (2.5-2.7) | (0.9-1.1) | (-2.91--2.42) |

*(Continued)*

| | 1990 Number | 2021 Number | 1990 | 2021 | Estimated Annual Percent Change |
|---|---|---|---|---|---|
| | (95% UI) | (95% UI) | Age-Standardized Rate | Age-Standardized Rate | (95% CI) |
| | | | (95% UI) | (95% UI) | |
| Southern Sub-Saharan Africa | 3035 | 3602.6 | 6.8 | 5.1 | -0.74 |
| | (2460.6-3528.5) | (3063.6-4183.9) | (5.7-8.0) | (4.4-5.9) | (-1.06--0.41) |
| Tropical Latin America | 1996.4 | 1365.2 | 1.57 | 0.56 | -3.41 |
| | (1910.7-2080.7) | (1286.1-1422.4) | (1.51-1.63) | (0.52-0.59) | (-3.64--3.18) |
| Western Europe | 4362.8 | 2549 | 0.94 | 0.32 | -3.49 |
| | (4186.9-4453.2) | (2288.1-2685.4) | (0.91-0.96) | (0.30-0.34) | (-3.62--3.35) |
| Western Sub-Saharan Africa | 7613.6 | 13860.4 | 4.1 | 3.9 | -0.24 |
| | (5165.4-9437.2) | (5501.6-19367.0) | (2.9-5.1) | (1.7-5.4) | (-0.55-0.06) |
| **Prevalence** | | | | | |
| Global | 99831015.3 | 111726710.3 | 2117.3 | 1334.9 | -1.54 |
| | (85229366.8-117584467.8) | (94066728.6-132527344.8) | (1800.7-2491.9) | (1125.8-1581.2) | (-1.62--1.47) |
| Andean Latin America | 1123702.3 | 1641285.4 | 3574.2 | 2493.5 | -1.23 |
| | (970028.1-1326322.2) | (1407113.6-1959525.9) | (3109.4-4176.7) | (2143.0-2967.1) | (-1.26--1.20) |
| Australasia | 944731.6 | 1252878.5 | 4311.6 | 3300.7 | -0.77 |
| | (791211.5-1141706.4) | (1045144.3-1504332.5) | (3601.9-5223.5) | (2753.4-3993.9) | (-0.83--0.72) |
| Caribbean | 1433681.6 | 1905115.9 | 4445.3 | 3786.8 | -0.34 |
| | (1223033.1-1720833.2) | (1632348.1-2276964.4) | (3813.5-5273.8) | (3242.7-4533.6) | (-0.54--0.25) |
| Central Asia | 2695748.1 | 2757244.4 | 4560.7 | 2881.1 | -1.74 |
| | (2354493.7-3134751.4) | (2361390.4-3237524.4) | (3978.3-5300.9) | (2467.0-3378.3) | (-1.87--1.62) |
| Central Europe | 4872211 | 3817589.4 | 3539.5 | 2494 | -1.13 |
| | (4146291.7-5724023.2) | (3187226.9-4553985) | (3013.1-4155.0) | (2091.6-2992.2) | (-1.18--1.08) |
| Central Latin America | 8032831.1 | 8468246.5 | 5560.4 | 3245.1 | -1.18 |
| | (6687527.9-10258155.8) | (7093267.2-10833930.7) | (4682.1-6998.6) | (2716.7-4154.8) | (-1.47--0.89) |
| Central Sub-Saharan Africa | 570964.8 | 1114769.6 | 1469.2 | 1120 | -0.9 |
| | (489901.5-678521.6) | (950923.5-1331821.3) | (1263.5-1719.5) | (956.8-1323.4) | (-1.01--0.80) |
| East Asia | 9475612.4 | 12711293.1 | 832.6 | 682.8 | -0.49 |
| | (7964928.3-11416953.9) | (10401874.7-15585652.6) | (696.2-1000.2) | (559.1-836.8) | (-0.61--0.36) |
| Eastern Europe | 9668474.8 | 7128084 | 3775 | 2574.9 | -1.56 |
| | (8161860.6-11349384.9) | (5977003.5-8349756.5) | (3200.6-4432.4) | (2149.8-3037.8) | (-1.93--1.19) |
| Eastern Sub-Saharan Africa | 3014221 | 5412902.6 | 2262.2 | 1709.3 | -0.98 |
| | (2588122.4-3605212.6) | (4624280.4-6485853.5) | (1947.2-2685.0) | (1462.2-2025.0) | (-1.07--0.91) |
| High-income Asia Pacific | 8156790.1 | 7439252.2 | 4183.8 | 2773.8 | -1.45 |
| | (6861570.9-9785543.9) | (6137812.4-8937164.7) | (3522.0-5016.8) | (2294.3-3366.1) | (-1.58--1.32) |
| High-income North America | 11189306.7 | 9809902.3 | 3557.1 | 2030.3 | -1.95 |
| | (9329779.3-13375637.2) | (8044282.1-11747273.8) | (2968.2-4251.3) | (1662.9-2442.7) | (-2.26--1.63) |
| North Africa and Middle East | 2525991 | 3626899.7 | 997.3 | 603.7 | -1.76 |
| | (2133929-2998895.8) | (3006184.2-4369208) | (834.8-1189.6) | (500.2-726.2) | (-1.88--1.63) |
| Oceania | 93082.7 | 231146.6 | 1868.4 | 1969.9 | 0.084 |
| | (80042.6-108372) | (200247.2-268735.2) | (1603.3-2177.2) | (1703.2-2293.3) | (-0.003-0.170) |
| South Asia | 6698291 | 10483440.9 | 784.1 | 589.3 | -1.04 |
| | (5616255.2-7966089) | (8721111.5-12516508.3) | (653.6-933.9) | (490.5-703.0) | (-1.19--0.88) |

*(Continued)*

**Table 2.** (Continued)

| | 1990 Number | 2021 Number | 1990 | 2021 | Estimated Annual Percent Change |
| --- | --- | --- | --- | --- | --- |
| | (95% UI) | (95% UI) | Age-Standardized Rate | Age-Standardized Rate | (95% CI) |
| | | | (95% UI) | (95% UI) | |
| Southeast Asia | 4430761.5 | 6792791.5 | 1170.5 | 925.3 | -0.65 |
| | (3790576.4-5300389.9) | (5700258.9-8112030.8) | (991.3-1390.0) | (777.3-1104.2) | (-0.73--0.57) |
| Southern Latin America | 3901569.3 | 4641010.3 | 8126.2 | 6115 | -0.77 |
| | (3389364.9-4561640.5) | (3989792-5447961.5) | (7067.0-9486.1) | (5244.9-7177.7) | (-0.90--0.63) |
| Southern Sub-Saharan Africa | 1135107 | 1428574.6 | 2862 | 1894.5 | -1.53 |
| | (969698.2-1340509.9) | (1213459.1-1693578.9) | (2414.5-3398.0) | (1605.7-2247.6) | (-1.67--1.38) |
| Tropical Latin America | 3740747.9 | 4015600.1 | 2863.5 | 1596.7 | -2.64 |
| | (3199129.3-4473054.5) | (3412595.5-4733456.7) | (2442.0-3360.4) | (1361.3-1887.4) | (-2.92--2.36) |
| Western Europe | 14034599.1 | 12677173.3 | 3119.9 | 2210.7 | -1.19 |
| | (11779739.2-16772001.7) | (10593100.1-15161649.8) | (2624.9-3743.0) | (1844.6-2663.0) | (-1.22--1.15) |
| Western Sub-Saharan Africa | 2092590.5 | 4371509.5 | 1485.2 | 1230.7 | -0.62 |
| | (1792687.6-2524018.7) | (3749007.4-5228113.8) | (1273.2-1766.1) | (1056.0-1449.8) | (-0.71--0.53) |

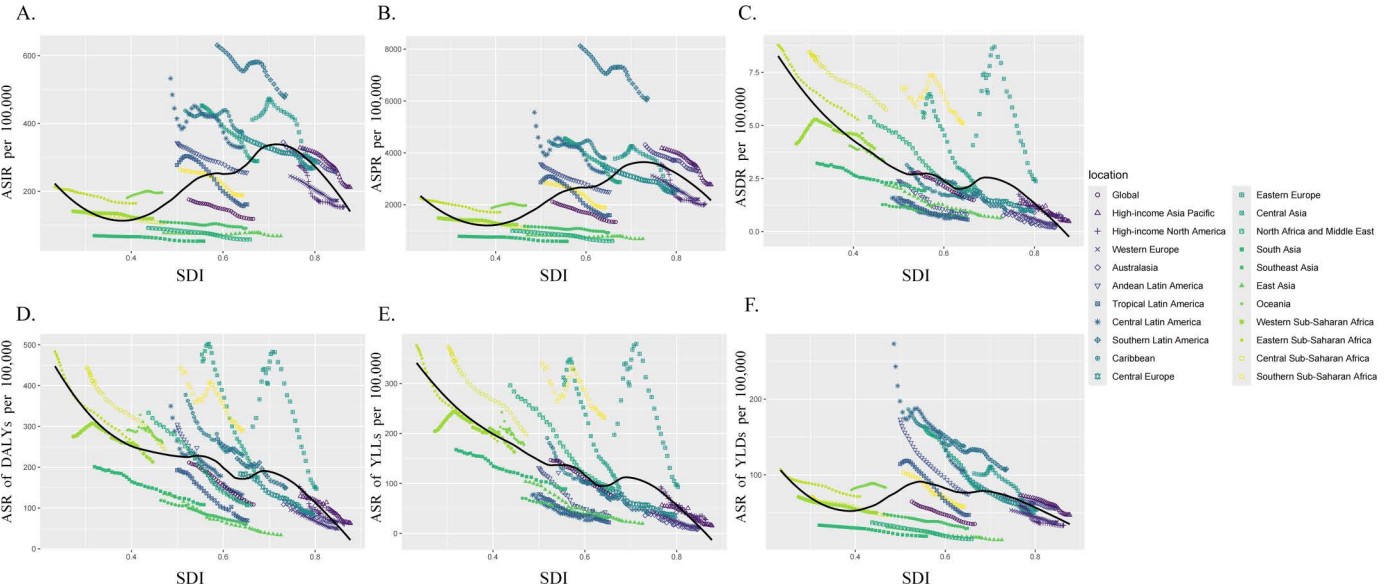

**Fig 2. Age-Standardized Rate of injuries caused by fire, heat and hot substances in 22 GBD regions with varying Social-Demographic Index globally from 1990 to 2021.** A. is for the Age-Standardized Rate of Incidence; B. is for the Age-Standardized Rate of Prevalence; C. is for the Age-Standardized Rate of Death; D. is for the Age-Standardized Rate of Disability-Adjusted Life Years. (E) The Age-Standardized Rate of Years of Life Lost; F. is for the Age-Standardized Rate of Years Lived with Disability.

Additionally, the Frontier analysis revealed that as societal and demographic development progresses, the EF have generally increased, indicating a greater potential for burden of burn improvement. Otherwise, the top five countries or regions with the greatest EF (42.14~72.99) were the Netherlands, Germany, Canada, the United States, and Iceland. Conversely, the top five countries or regions with the smallest EF (0~4.09) were Somalia, Timor-Leste, Papua New Guinea, Yemen, and Laos ((Figs 3, S3 File).

**Forecasting burn injuries burden from 2021 to 2030 base on BAPC model**

Due to the direct influence of data volume on the fitting error of the BAPC model, this study selected DALYs, Prevalence and Death, which have larger data volumes, as the indicators for BAPC model forecasting.

The goodness of fit of the ASR forecast fan plots for both sexes, as depicted in Figs 4, was highly satisfactory for DALYs (Figs 4A, 4B), Prevalence (Figs 4C, 4D), and Death (Figs 4E, 4F). The precision of the forecasts was indicated by the narrowness of the fan plot sectors, signifying a reduced level of uncertainty and an optimal model fit. Furthermore, the projections extended to the year 2030, revealing a consistent downward trajectory in the burden of burn injuries for both sexes.

As we had not only predicted the ASR, but also the Number worldwide. The results of BAPC indicated that although the Number of DALYs for children were projected to decrease over the next 9 years, they remain high across all age groups. Additionally, each indicator had a peak value. Furthermore, by 2030, the Number of DALYs were projected to decrease for both sexes. However, males Prevalence was projected to increase from 5,476,178 cases in 2021–6,012,334 cases in 2030, while female from 5,694,956 cases to 6,269,209 cases totally. Lastly, as shown in S4 File, Death for both sexes were increasing in age groups above 70; although there is a decrease by 2030, the Number remained relatively high, with the peak focusing on the 0–9 age group (S4 File).

In addition, the predictive outcomes of the BAPC model indicated that by 2030, the ASR of DALYs for females and males are projected to decrease from 114.09 and 90.83 per 100,000 population in 2021 to 92.89 and 77.62 per 100,000 population, respectively, representing an approximate 20% reduction for both sexes. Furthermore, the ASPR for females and males had also been projected to decline from 1333.41 and 1310.82 per 100,000 population in 2021 to 1281.93 and 1256.62 per 100,000 population by 2030, marking a reduction of about 4% for both sexes. Lastly, the ASDR for females and males were anticipated to drop from 1.49 and 1.34 per 100,000 population in 2021 to 1.19 and 1.10 per 100,000 population by 2030, which corresponds to a reduction of approximately 20% for both sexes. Moreover, the ASR of burn injuries across different age groups revealed that the current severe burden of burn injuries is predominantly concentrated among the elderly population (S4 File).

## Discussion

Overall, the GBD from these injuries shows a declining trend. While some Numbers have increased, ASR have decreased. This may be linked to the advancements in contemporary medical care and the increasing awareness of disease prevention. With technological progress in burn injuries care, rehabilitation medicine, and other related fields, treatments such as autologous and allogenic skin grafting, management of severe infections, emergency response, and functional reconstruction during disease stability can contribute to reducing the disease burden of it [24,25].

Burn injuries, for instance, result from multiple causes such as contact with flames, smoke, hot materials, and electrical injuries. Patients with burn injuries face a range of complications including physical and psychological impairments, sepsis, among others [3]. Based on the research findings of disease burden studies across different regions with varying SDI levels, it is evident that regions with heavier burn injuries burden are predominantly concentrated in middle to high SDI regions. For instance, a retrospective study conducted in Victoria, Australia from 2003 to 2021 had revealed a declining trend in hospitalization rates due to occupational exposure to fires, flames, or smoke. However, injuries caused by occupational exposures were found to be more severe compared to non-occupational injuries [26]. Simultaneously, a population-based study conducted in the area from 2008 to 2017 identified a higher disease burden of burn injuries among male patients, consistent with the findings of S2 File in this study comparing disease burdens between male and female in Australia [27]. In addition to high SDI regions, significant sex differences in GBD are also observed in middle SDI regions. Taking South Asia in S2 File as an example, the DALYs and Prevalence among females were much higher than males. Sex differences arise from multiple factors. This may be attributed to the widespread industrialization and advanced technology in high-SDI regions and countries, where males are more commonly employed in mechanical

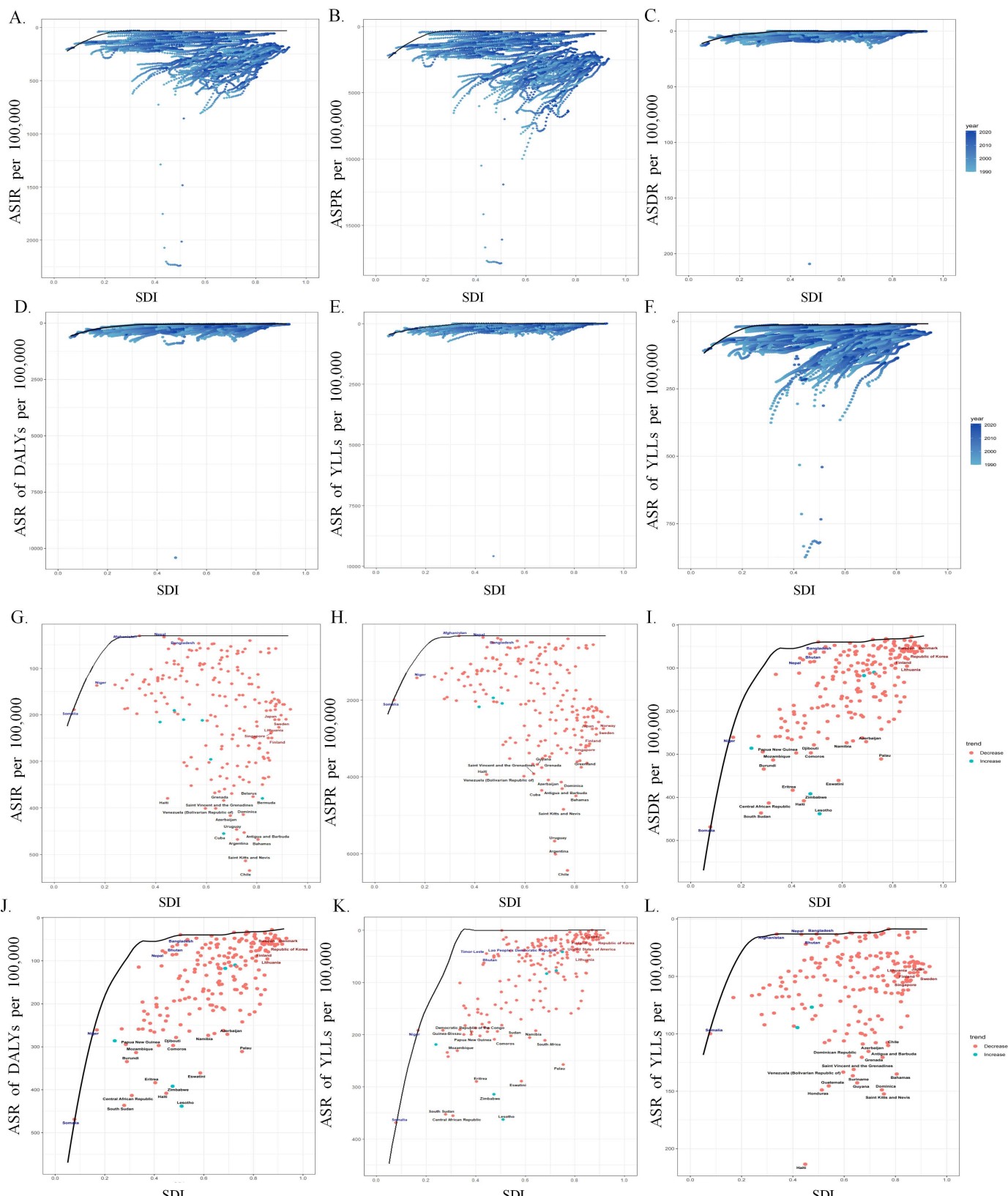

**Fig 3. Frontier analysis of injuries' burden caused by fire, heat, and hot substances from 1990 to 2021.** Frontier analysis is depicted with a black solid line representing the boundary, and points representing countries or regions. The trends of Social-Demographic Index and Age-Standardized Rate

for each country and region are presented in a gradient of blue. And the 15 countries with the largest effective differences are marked in black font. Countries with lower SDI (<0.5) and lower EF are marked in blue font. Countries or regions with higher SDI (>0.8) and higher EF are marked in red font. Additionally, from 1990 to 2021, red points indicate an increase in the Age-Standardized Rate of burn; blue points indicate a decrease in the Age-Standardized Rate of burn. **A.** - F. represent the burden changes of burn Incidence, Prevalence, Death, Disability-Adjusted Life Years, Years of Life Lost, and Years Lived with Disability in each country or region from 1990 to 2021, as well as the distance to the "Frontier." G. - L. represent the burden of burn Incidence, Prevalence, Death, Disability-Adjusted Life Years, Years of Life Lost, and Years Lived with Disability in each country or region in 2021, as well as the distance to the "Frontier.".

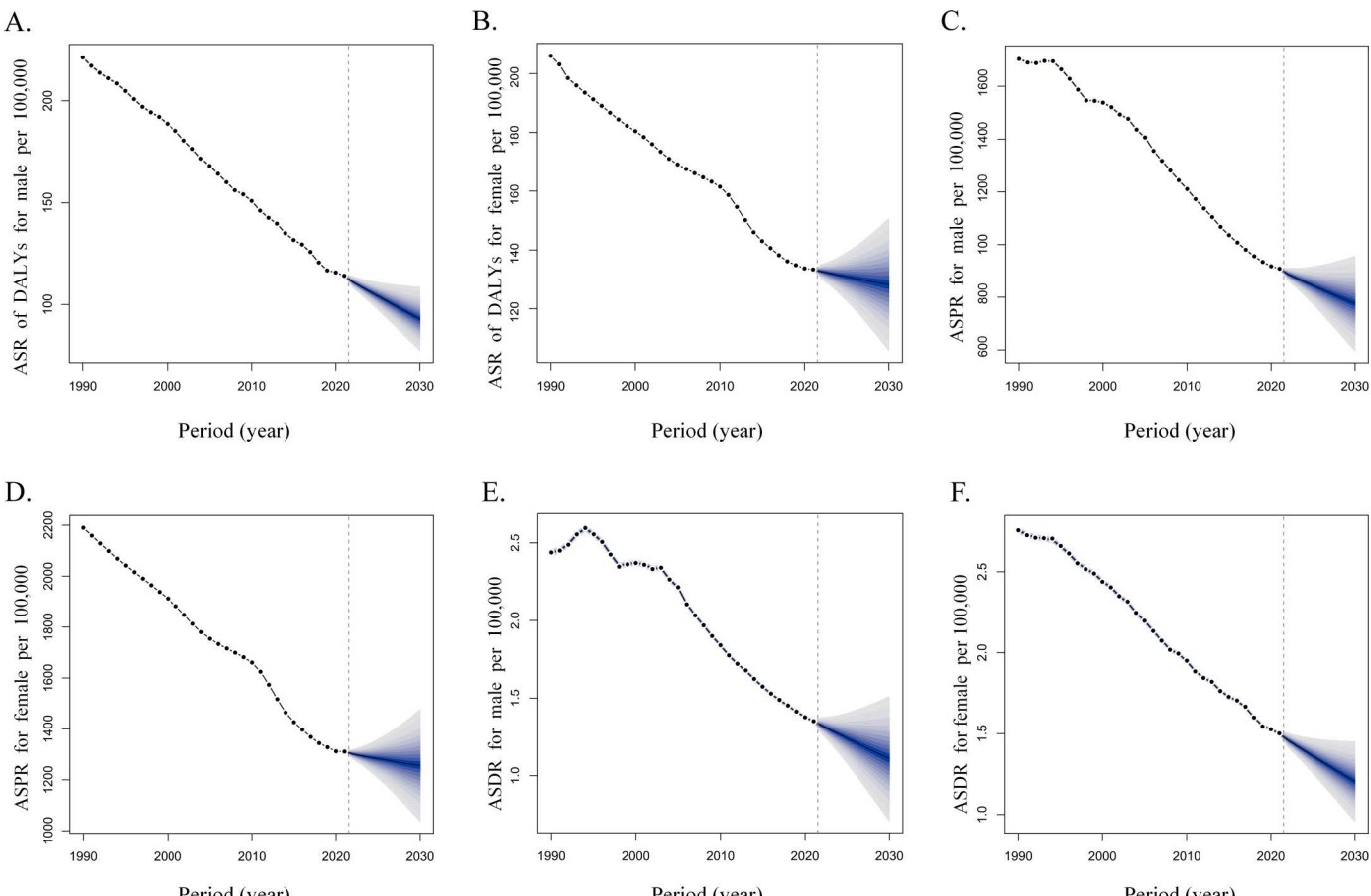

**Fig 4. Predictions of Age-Standardized Rate of burden caused by fire, heat and hot substances for different sexes in 2030 based on the Bayesian Age-Period-Cohort model.** A. and B. is for the prediction of the male and female Age-Standardized Disability-Adjusted Life Years Rate respectively; B. is for the prediction of the female Age-Standardized Disability-Adjusted Life Years Rate; C. and D. is for the prediction of male and female Age-Standardized Prevalence Rate; E. and F. is for the prediction of male and female Age-Standardized Death Rate.

engineering and jobs involving exposure to hot substances. In contrast, in regions with lower economic levels, cooking safety is often compromised, contributing to different patterns in burn injuries incidence between sexes. Concurrently, there is supporting evidence for our findings. A systematic review based on South Asia indicated that flame-related burn injuries and scalds accounted for over 80% of total burn injuries in the region, being the most common types of injuries observed in children and women [28]. Furthermore, predictions from the BAPC model suggest that children aged 0–9 are at extremely high risk. Several studies indicate that unintentional contact with heat sources by children is often a key contributing factor to the persistent occurrence of childhood burn injuries cases [29–32].

Overall, the GBD of burn injuries had decreased. This can be attributed to advancements in disease treatment and functional recovery. Developments such as autologous skin grafting, adipose-derived stem cell in situ injection, pressure garments, local silicone therapy, and burn physical therapy have significantly alleviated the disease burden among burn injuries survivors [33–37]. But clinical and rehabilitation treatments for burn injuries patients still face challenges. This is evident in the obstacles burn injuries survivors encounter when returning to work, with reports indicating that fewer than half of the included participants regained productivity (46%). Various complex factors contribute to this, including ages, burn injuries severity, and post-injury employment status [38].This undoubtedly confirms the substantial global economic burden imposed by burn injuries [6]. Additionally, patients with extensive burn injuries face a high probability of sepsis and have limited availability of autologous skin. Consequently, autologous skin grafting and multidrug-resistant bacterial infections are often challenging issues for patients. However, novel treatment approaches for burn injuries patients continue to emerge, such as cultured epidermal autografts applicable to patients with extensive burn and limited autologous skin survival, as well as mitochondrial transplant therapy [39,40]. However, such emerging treatment approaches are still in their nascent stages, requiring further research to explore and refine their efficacy.

In this study, we using the BAPC model to forecast number from 2021 to 2030 by sexes, the elderly population shows a continued increase in Death compared to 2021 (Figs 4). Therefore, in the context of a decreasing ASR of disease burden caused by hot, heat and hot substances, policymakers should proactively implement measures to reduce the likelihood of an increase in number. Such measures could include improving summer cooling conditions, increasing vegetation, conducting safety education, and enhancing public awareness [41].

Lastly, Frontier analysis plays a crucial role in optimizing the allocation of health resources. It can assist policymakers in prioritizing efficient health interventions within limited resources to achieve the greatest health benefits. By identifying the best performers among efficiency Frontier countries, targeted policies can be formulated to optimize the investment of funds and resources. Specifically, by comparing the performance of different countries at the same SDI level, we can learn from the successful experiences of efficiency frontier countries and develop more effective disease control and prevention policies. The frontier analysis in this study reveals that the potential for burn burden improvement is lower in low SDI countries and higher in high SDI countries. This insight underscores the importance of tailored interventions that consider the specific development contexts of different regions. By leveraging the knowledge of best practices from countries on the efficiency frontier, policymakers can make more informed decisions and allocate resources strategically to address the health challenges that are most pertinent to their own socio-demographic context [17–19,42].

This study conducted multidimensional analyses across different sexes, age groups, and levels of SDI to understand GBD trends. However, this study also has limitations, unlike epidemiological studies focusing on specific regions or single-center studies, GBD research cannot achieve precise analysis. Subsequently, the GBD 2021 data does not differentiate the severity of injuries caused by fire, heat, and hot substances. Moreover, due to variations in the efficiency of healthcare and public health statistical systems across different SDI regions, data collected from regions with lower efficiency may exhibit disparities. The aforementioned deficiencies may lead to inaccuracies in the current global burden of disease estimates for burn.

## Conclusion

Both the current and predicted results indicated an increasing burden in the Number of injuries caused by fire, heat and hot substances, and the ASR showed a declining trend globally, especially in lower SDI regions. And this study assists policy makers to monitor the global burden of burn. Moreover, it can also provide solid scientific evidence for local health authorities and burn surgical professionals to conduct evidence-based interventions to prevent its development among the identified vulnerable populations.

## Supporting information

**S1 Fig. Age-Standardized Rate of injuries caused by fire, heat and hot substances in 204 countries with varying Social-Demographic Index globally in 2021.** A. is for the Age-Standardized Rate of Incidence; B. is for the Age-Standardized Rate of Prevalence; C. is for the Age-Standardized Rate of Death; D. is for the Age-Standardized Rate of Disability-Adjusted Life Years. (E) The Age-Standardized Rate of Years of Life Lost; F. is for the Age-Standardized Rate of Years Lived with Disability.
(PDF)

**S1 File. Formulas of disease burden indicators.** Formulas of Age-Standardized Rate, Estimated Annual Percentage Change, Frontier Analysis, and Bayesian Age - Period - Cohort model.
(DOCX)

**S2 File. The burden of injuries caused by fire, heat and hot substances in female and male across 22 GBD regions.** The burden of Disability-Adjusted Life Years, Years of Life Lost, Years Lived with Disability, Incidence, Death, and Prevalence of burn in male and female.
(XLSX)

**S3 File. Frontier Analysis: Countries with high potential for improvement in the burden of injuries caused by fire, heat, and hot substances in 2021.** The burden of Disability-Adjusted Life Years, Years of Life Lost, Years Lived with Disability, Incidence, Death, and Prevalence of burn in high potential for improvement.
(XLSX)

**S4 File. The burden forecast of injuries caused by fire, heat, and hot substances for 2021–2030.** The forecast of Prevalence, Death, and Disability-Adjusted Life Years Numbers as well as Age-Standardized Rates for injuries caused by fire, heat, and hot substances in males and females from 2021 to 2030.
(XLSX)

## Acknowledgments

This study most wants to thank the GBD2021 collaborators, because without their summary, there would be no disease burden data for so many diseases.

## Author contributions

**Conceptualization:** Shi Huang, Hui-Zhen Lin, Xin Wei.

**Data curation:** Shi Huang, Hui-Zhen Lin.

**Formal analysis:** Shi Huang.

**Funding acquisition:** Shi Huang.

**Investigation:** Shi Huang, Hui-Zhen Lin.

**Methodology:** Shi Huang, Hui-Zhen Lin, Xin Wei.

**Project administration:** Shi Huang, Hui-Zhen Lin.

**Resources:** Shi Huang.

**Software:** Shi Huang, Xin Wei.

**Supervision:** Shi Huang, Xin Wei.

**Validation:** Shi Huang, Xin Wei.

**Visualization:** Shi Huang, Xin Wei.

**Writing – original draft:** Shi Huang.

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
