## [Decision Letter · Decision Letter 0]

26 Mar 2025

Dear Dr. Huang,

Thank you for submitting your manuscript to PLOS ONE. After careful consideration, we feel that it has merit but does not fully meet PLOS ONE’s publication criteria as it currently stands. Therefore, we invite you to submit a revised version of the manuscript that addresses the points raised during the review process.

There are significant issues pointed out by the reviewers that need to be addressed. 

We look forward to receiving your revised manuscript.

Kind regards,

David G. Greenhalgh, MD

Academic Editor

PLOS ONE

Journal Requirements:

Self-funded research project of Health and Family Planning Commission of Guangxi Zhuang Autonomous Region(Z20180240)

4. Please include a copy of Table 1 and 2, which you refer to in your text on page 5.

5. Please remove all personal information, ensure that the data shared are in accordance with participant consent, and re-upload a fully anonymized data set.

Additional Editor Comments :

The reviewers have serious concerns about the paper. If you would like to address the concerns, please respond to the critiques.

Reviewers' comments:

Reviewer's Responses to Questions

**Comments to the Author**

1. Is the manuscript technically sound, and do the data support the conclusions?

Reviewer #1: Partly

Reviewer #2: Yes

2. Has the statistical analysis been performed appropriately and rigorously?

Reviewer #1: I Don't Know

Reviewer #2: Yes

3. Have the authors made all data underlying the findings in their manuscript fully available?

Reviewer #1: Yes

Reviewer #2: Yes

4. Is the manuscript presented in an intelligible fashion and written in standard English?

Reviewer #1: No

Reviewer #2: Yes

Reviewer #1: Thank you for allowing me the opportunity to review the manuscript entitled: Global, regional, national burden of injuries cause by fire, heat, and hot substances from 1990 to 2021. I do not have experience in statistical analysis and will leave that to the Editor and other reviewers sourced.

Statements such as this one are questionable and would need support from the surrounding literature “Burn injuries, is thought to be a chronic disease of the injury, typically result from contact with fire, heat, and hot substances”.

The authors state the following but only provide one reference despite stating studies: “Recent epidemiological studies on burn injury have also proliferated. In terms of burns injury, studies indicate that in 2019, the global economic burden attributable to burns amounted to USD 11.2 billion, accounting for 0.09% of the global Gross Domestic Product (GDP)7.

The manuscript requires review of syntax and grammar throughout as the flow is disjointed.

The manuscript states in the introduction that the study aims to estimate the overall GBD of burn injury using the GBD 2021, and seeks to better guide global public health policymakers in the field of burn injury. However, in the methods the aims state to describe the global burden of disease caused by injuries from fire, heat, and hot substances using data from the GBD 2021 database. Then in the discussion it states “This study explores the GBD attributed to injuries caused by hot, heat, and hot substances from 1990 to 2021, and forecasts from 2021 to 2030. Therefore, there is inconsistency in what the actual aim of the study is.

There are no limitations section in the discussion that discusses the limitations and the conclusion is limited.

Reviewer #2: Abstract

The conclusions and results sections overlap considerably, with no clear distinction between the two.

Main Text

The manuscript estimates the overall global burden of burn injury using GBD 2021 data and subsequently applies a Bayesian Age-Period-Cohort (BAPC) model to forecast the burden of burn injuries from 2021 to 2030. A major limitation is the absence of more recent data beyond 2021. The inclusion of data from 2021 to 2024 would significantly enhance the analysis, allowing a comparison between predicted and actual trends over this period.

Additionally, the introduction includes several references to therapeutic interventions (e.g., post-graft hyperbaric oxygen treatment) that are not directly relevant to the study's primary objective. This detracts from the focus and scientific coherence of the paper.

The excessive use of abbreviations and a large number of figures and data included in the supplementary material significantly impairs the readability of the manuscript.

I strongly recommend a thorough language review and editing of the manuscript to improve its clarity and academic rigor.

**Do you want your identity to be public for this peer review?** For information about this choice, including consent withdrawal, please see our Privacy Policy

Reviewer #1: No

Reviewer #2: No

---

## [Author Response · Author response to Decision Letter 0]

9 Apr 2025

Dear editors and reviewers,

We appreciate the valuable comments of the editors and reviewers. Our Response to Reviewers are as follows.

Notes from the editor:

Response: Thank you very much for your thorough review and valuable feedback on our manuscript. We sincerely appreciate your comments regarding the need for revisions in the manuscript format. We have made the necessary adjustments according to the title page and main text format provided by the editor. Once again, we are grateful for your suggestions on the manuscript format, which will help us better align with the style of PLOS ONE.

2. Thank you for stating the following financial disclosure: Self-funded research project of Health and Family Planning Commission of Guangxi Zhuang Autonomous Region(Z20180240)Please state what role the funders took in the study. If the funders had no role, please state: "The funders had no role in study design, data collection and analysis, decision to publish, or preparation of the manuscript. "If this statement is not correct you must amend it as needed. Please include this amended Role of Funder statement in your cover letter; we will change the online submission form on your behalf.

Response: Thank you very much for your thorough review and insightful comments on our manuscript. We have revised the financial disclosure as suggested by the editor and added the statement "The funders had no role in study design, data collection and analysis, decision to publish, or preparation of the manuscript." in lines 397-400. Once again, we are grateful for the editor's suggestions, which have helped to enhance the rigor of our research.

Response: Thank you very much for your thorough review and valuable feedback. We are truly grateful for your guidance regarding the need for an ethical statement. We have removed the ethical statement that was placed after the main text and have instead supplemented the Materials and methods section with the ethical statement for our study in lines 88-92. Once again, we extend our sincere thanks for the editor’s insightful suggestions.

4. Please include a copy of Table 1 and 2, which you refer to in your text on page 5.

Response: We thank the editor for carefully reviewing our work. We will submit copies of Table 1 and Table 2 along with the revised manuscript. Once again, we express our sincere gratitude for the editor's valuable comments!

5. Please remove all personal information, ensure that the data shared are in accordance with participant consent, and re-upload a fully anonymized data set. Note: spreadsheet columns with personal information must be removed and not hidden as all hidden columns will appear in the published file. Additional guidance on preparing raw data for publication can be found in our Data Policy (https://journals.plos.org/plosone/s/data-availability#loc-human-research-participant-data-and-other-sensitive-data) and in the following article: http://www.bmj.com/content/340/bmj.c181.long.

Response: Thank you very much for your thorough review and insightful comments. Our study does not involve any personal information of participants. All data used in this study are sourced from the Global Burden of Disease Study 2021 (GBD 2021) database, which is a publicly available and open-access database. All the data utilized in our research can be freely and publicly downloaded from the website [Global Burden of Disease Study 2021 (GBD 2021) Data Resources] (https://ghdx.healthdata.org/gbd-2021).

Response: Thank you very much for your meticulous observation and valuable suggestions. We have revised the Supporting information description and updated the citation in lines 555-576 according to the format you provided. Once again, we sincerely appreciate your insightful comments.

Reviewer #1

1. Statements such as this one are questionable and would need support from the surrounding literature “Burn injuries, is thought to be a chronic disease of the injury, typically result from contact with fire, heat, and hot substances”.

Response: Thank you very much for your valuable comments. After carefully re-reading the relevant papers, we realized that our original statement was problematic, for which we sincerely apologize. Therefore, we have revised the original statement to “Burn injuries are inherently acute conditions; however, in certain circumstances, the long-term sequelae of burn injuries may exhibit characteristics akin to those of chronic diseases,” based on the literature, and added the references (lines 36-38). Once again, we express our sincere gratitude for the crucial suggestions from the reviewer, which have enhanced the scientific rigor of our study.

2. The authors state the following but only provide one reference despite stating studies: “Recent epidemiological studies on burn injury have also proliferated. In terms of burns injury, studies indicate that in 2019, the global economic burden attributable to burns amounted to USD 11.2 billion, accounting for 0.09% of the global Gross Domestic Product (GDP)7.

Response: Thank you very much for your meticulous observation and critical comments. Upon our review, we realized that “studies” should be “study” in this context, and we have reflected deeply on this mistake. Accordingly, we have made the necessary revisions in lines 46-48 of the manuscript: “A recent epidemiological study indicates that in 2019, the global economic burden attributable to burns amounted to USD 11.2 billion, accounting for 0.09% of the global Gross Domestic Product (GDP) [6].” Once again, we sincerely appreciate your valuable suggestions.

3. The manuscript requires review of syntax and grammar throughout as the flow is disjointed.

Response: Thank you very much for taking the time to read our manuscript. After a careful re-examination, we have indeed found several linguistic issues. We have made every effort to revise the wording and grammar. Should there still be any oversights, please do point them out. We truly appreciate your valuable feedback!

4. The manuscript states in the introduction that the study aims to estimate the overall GBD of burn injury using the GBD 2021, and seeks to better guide global public health policymakers in the field of burn injury. However, in the methods the aims state to describe the global burden of disease caused by injuries from fire, heat, and hot substances using data from the GBD 2021 database. Then in the discussion it states “This study explores the GBD attributed to injuries caused by hot, heat, and hot substances from 1990 to 2021, and forecasts from 2021 to 2030. Therefore, there is inconsistency in what the actual aim of the study is.

Response: Thank you very much for your critical comments. After our review, we have indeed found that there were several confusing statements regarding the study aims throughout the manuscript, for which we have conducted a self-reflection. Therefore, we have retained only the statement of the study aims in lines 65-71 of the Introduction, and removed or streamlined the rest. Once again, we sincerely appreciate your valuable suggestions, which have enhanced the scientific rigor of our study.

5. There are no limitations section in the discussion that discusses the limitations and the conclusion is limited.

Response: Thank you very much for your valuable suggestions! Regarding the limitations section of the manuscript, which can be found in lines 374-381 of the revised manuscript. As for the issues in the conclusion, we have conducted a profound self-reflection and have rewritten the conclusion in lines 383-389. Once again, we sincerely appreciate your insightful comments, which have greatly enhanced the rigor of our manuscript.

Reviewer #2

1. The conclusions and results sections overlap considerably, with no clear distinction between the two.

Response: Thank you very much for the time and effort you have dedicated to reviewing our study. We have carefully revised the results and conclusion sections in the abstract and the main text to address the repetition and other oversights. The revisions can be found in lines 24-31 and 383-389. Thank you again for your insightful feedback.

2. The manuscript estimates the overall global burden of burn injury using GBD 2021 data and subsequently applies a Bayesian Age-Period-Cohort (BAPC) model to forecast the burden of burn injuries from 2021 to 2030. A major limitation is the absence of more recent data beyond 2021. The inclusion of data from 2021 to 2024 would significantly enhance the analysis, allowing a comparison between predicted and actual trends over this period.

Response: Thank you very much for your critical comments. The latest version of the Global Burden of Disease (GBD) database is the 2021 edition, which is updated every two years. Therefore, the most recent burn injuries burden data available are from 1990 to 2021. In our study the latest and most complete burden of disease data on burns from the GBD has been used.

3. Additionally, the introduction includes several references to therapeutic interventions (e.g., post-graft hyperbaric oxygen treatment) that are not directly relevant to the study's primary objective. This detracts from the focus and scientific coherence of the paper.

Response: Thank you very much for your valuable suggestions. Regarding the overemphasis on burn injuries interventions and treatment methods, as well as the excessive references cited in this area, which you pointed out, we have reflected on this issue and agree with your observation. Therefore, we have streamlined the introduction of burn injuries treatment methods (lines 41-44) to better align the study with its core focus. Once again, we sincerely appreciate your insightful comments, which have significantly enhanced the scientific rigor of our manuscript.

4. The excessive use of abbreviations and a large number of figures and data included in the supplementary material significantly impairs the readability of the manuscript.

Response: Thank you very much for your valuable suggestions. Upon reflection, we have realized that the excessive number of tables and figures in our Supporting information did indeed affect readability. Therefore, we have removed the redundant Supporting information 3 and presented all necessary Supporting information in a standardized format, which has significantly improved readability. Once again, we sincerely appreciate your comments, which have greatly enhanced the readability of our manuscript.

5. I strongly recommend a thorough language review and editing of the manuscript to improve its clarity and academic rigor.

Response: Thank you very much for taking the time to read our manuscript. After a careful re-examination, we have indeed found some linguistic issues. We have made every effort to revise the wording and grammar. Should there still be any oversights, please do point them out. We truly appreciate your valuable feedback!

We thank the editors and reviewers for their valuable review time and constructive comments. We have improved the manuscript on an item-by-item basis according to the feedback. If there are still any adjustments that need to be corrected, we would be grateful for further corrections.

Shi Huang

Affiliated Wuming Hospital of Guangxi Medical University

huangshiwuming@163.com

---

## [Decision Letter · Decision Letter 1]

28 Apr 2025

Global, regional, national burden of injuries caused by fire, heat, and hot substances from 1990 to 2021

PONE-D-25-06429R1

Dear Dr. Huang,

We’re pleased to inform you that your manuscript has been judged scientifically suitable for publication and will be formally accepted for publication once it meets all outstanding technical requirements.

Kind regards,

David G. Greenhalgh, MD

Academic Editor

PLOS ONE

Additional Editor Comments (optional):

Accept

Reviewers' comments:

Reviewer's Responses to Questions

**Comments to the Author**

Reviewer #1: All comments have been addressed

Reviewer #2: All comments have been addressed

2. Is the manuscript technically sound, and do the data support the conclusions?

Reviewer #1: Yes

Reviewer #2: Yes

3. Has the statistical analysis been performed appropriately and rigorously?

Reviewer #1: I Don't Know

Reviewer #2: Yes

4. Have the authors made all data underlying the findings in their manuscript fully available?

Reviewer #1: Yes

Reviewer #2: Yes

5. Is the manuscript presented in an intelligible fashion and written in standard English?

Reviewer #1: Yes

Reviewer #2: Yes

Reviewer #1: Thank you for allowing me to review once again the manuscript entitled: Global, regional, national burden of injuries caused by fire, heat, and hot substances from 1990 to 2021. Thank you for taking the time to addressing the previous comments accordingly. The manuscript reads better after the revisions were addressed.

Reviewer #2: The authors have adequately addressed my comments, and I recommend this manuscript for publication

**Do you want your identity to be public for this peer review?** For information about this choice, including consent withdrawal, please see our Privacy Policy

Reviewer #1: No

Reviewer #2: No

---

## [Editor Report · Acceptance letter]

PONE-D-25-06429R1

PLOS ONE

Dear Dr. Huang,

I'm pleased to inform you that your manuscript has been deemed suitable for publication in PLOS ONE. Congratulations! Your manuscript is now being handed over to our production team.

Kind regards,

on behalf of

Dr. David G. Greenhalgh

Academic Editor

PLOS ONE